# Reduction in Atmospheric Particulate Matter by Green Hedges in a Wind Tunnel

**Marcello Biocca *** , **Daniele Pochi, Giancarlo Imperi and Pietro Gallo**

Consiglio per la Ricerca in Agricoltura e l'Analisi dell'Economia Agraria—CREA, Centro di Ricerca Ingegneria e Trasformazioni Agroalimentari, Via della Pascolare 16, 00015 Monterotondo, Italy; daniele.pochi@crea.gov.it (D.P.); giancarlo.imperi@crea.gov.it (G.I.); pietro.gallo@crea.gov.it (P.G.)
* Correspondence: marcello.biocca@crea.gov.it

**Abstract:** Urban vegetation plays a crucial role in reducing atmospheric particulate matter (PM), modifying microclimates, and improving air quality. This study investigates the impact of a laurel hedge (*Laurus nobilis* L.) on airborne PM, specifically total suspended particulate (TSP) and respirable particles ($PM_4$) generated by a Diesel tractor engine. Conducted in a wind tunnel of approximately 20 m, the research provides insights into dust deposition under near-real-world conditions, marking, to our knowledge, the first exploration in a wind tunnel of this scale. Potted laurel plants, standing around 2.5 m tall, were arranged to create barriers of three different densities, and air dust concentrations were detected at 1, 4, 9, and 14 m from the plants. The study aimed both to develop an experimental system and to assess the laurel hedge's ability to reduce atmospheric PM. Results show an overall reduction in air PM concentrations (up to 39%) due to the presence of the hedge. The highest value of dust reduction on respirable particles was caused by the thickest hedge (three rows of plants). However, the data exhibit varying correlations with hedge density. This study provides empirical findings regarding the interaction between dust and vegetation, offering insights for designing effective hedge combinations in terms of size and porosity to mitigate airborne particulate matter.

**Keywords:** urban forestry; air pollution; PM capture; particle deposition; porosity

## 1. Introduction

The impressive urban and landscape transformations of recent decades have corresponded to a dramatic growth of the urban population. Currently, more than half of the world's population (56.2%) lives in urban areas—progressively more in highly-populated cities [1]. In Italy, 69.5% of the population is urban [2]. Sustainable management of cities is a crucial point in stimulating the global ecological transition and mitigating climate change.

A major environmental health concern in urban areas is air pollution. The reduction in atmospheric pollutants and airborne particulate matter (PM) (or atmospheric dust) is one of the positive effects that urban vegetation provides to improve the quality of life. The term atmospheric dust refers to a mixture of solid and liquid particles suspended in the air which vary in size, composition, and origin. Some of the particles that make up atmospheric dust are emitted by various natural and anthropogenic sources ("primary particles"), while others derive from a series of chemical and physical reactions that occur in the atmosphere ("secondary particles"). There are also various removal mechanisms that the dust undergoes, including mechanisms that "remove" the dust from the atmospheric environment by causing it to fall back to the ground or towards the aquatic environment, as well as deposition mechanisms and active or passive obstacles for their interception [3].

The basis of dust classification is made up of the diameter of the particles (directly related to the possible interactions with the human body and, more specifically, with the respiratory and cardiovascular system) and their concentration. $PM_{10}$ (with an average aerodynamic diameter of less than 10 µm) represents the dust capable of penetrating the

upper respiratory tract, while $PM_{2.5}$ represents the dust able to penetrate the lower tract of the respiratory system (lung alveoli), causing serious harm to human health [4,5]. The health and quality of life of people living in cities are seriously threatened by air pollution. It is estimated that, in the year 2020, there were 52,300 premature deaths in Italy due to air exposure to $PM_{2.5}$ [6]. All over the world, the $PM_{2.5}$-related global premature mortality is estimated to be around 4.3 and 4.4 million [7].

Several measures can be set up to reduce particulate matter emissions and mitigate their effects, including the presence of vegetation, which plays a key role in contrasting the air pollution by means of different mechanisms. Numerous studies have shown that the plants offer an effective method of air purification, especially for non-point source particulate pollution, because of their wide distribution [8,9].

At least three major types of green infrastructure contribute to this effect: trees and hedges (which make up the so-called vertical greenery), surfaces (such as turf), and green roofs and walls. In particular, hedges represent an important and widespread form of greenery in the city. Although they vary greatly in terms of the species used, vegetative habitus, size, and density of foliage, hedges have a common trait: they are generally located in proximity to the main sources of city pollution, such as vehicular traffic. Therefore, hedges can be considered the first line of defense against smog emissions resulting from vehicular traffic. Hedges are also characterized by having a rather homogeneous morphology, with continuous vegetation from the base to the apex of the crown and a shape that can be traced back to a parallelepiped or a cuboid.

The main reason for the presence of hedges in urban environments lies in their ability to function as visual barriers. For example, hedges are placed in traffic reservations to reduce the nuisance of glare caused by vehicles passing in the opposite lane, around gardens and houses to increase privacy, and around sports facilities to help athletes focus. Their great aesthetic function is undeniable, including their presence in historic gardens and parks, where they may represent a distinctive feature, as seen in formal or Italian gardens. Together with their aesthetic functions, hedges contribute to the urban green endowment and the supply of ecosystem services typical of urban greenery, such as mitigation of the heat island effect [10], reduction in noise pollution [11,12], increase in biodiversity [13], and improvement of the microclimate [14]. Authors have even studied the role of urban hedges in blast mitigation [15]. Hedges are also attractive as they contribute to the resilience and adaptability of urban forests due the fact that they increase structural diversity in terms of age, spatial profile, and species distribution. Structural (and, thus, functional) diversity can be achieved quickly by planting appropriate shrubs in addition to slower-growing trees [16].

It is worth noting that hedges, unlike tree cover, can reduce air pollution even in places characterized by the presence of tall buildings around relatively narrow roads (also known as "road canyons"), where air ventilation and dilution of pollutants may be insufficient [17]. In these situations, the cover of tree crowns may sometimes cause a local accumulation of dust and pollutants, while the presence of hedges does not have this effect. Results have indicated that different kinds of greenbelts can improve footpath air quality to a certain degree (7–15%). Interestingly, the vegetation structure of shrubs and small trees (about 2–4.5 m in height) with small crown diameters shows the highest $PM_{10}$ removal efficiency along major or heavy traffic roads [18,19]. In open road locations, most studies report a reduction in the concentrations of various airborne pollutants due to hedgerows in the range of 15–60% [20]. Other authors have assessed the pollutant removal capacity in relation to botanical species [21,22]. Moreover, the habitus of the species (evergreen or broadleaves) influences the total yearly capacity of dust interception, which is stronger in evergreen than in broad-leaved plants.

Our research focused on the ability of hedges to remove particulate matter or atmospheric dust. Many authors have studied removal mechanisms and assessed the capacity of green barriers to reduce particulate matter using different techniques. Experimental methods include field experiments, chamber experiments, modelling and computer sim-

ulations, remote sensing, and wind tunnels. Wind tunnels are experimental setups that simulate the flow of air in a controlled environment over a specific area. Generally, air containing particulate matter is introduced into the tunnel and passed over a section of vegetation. The air is then measured for the concentration of PM both before and after it has passed through the vegetation to quantify the amount of PM that has been removed by the vegetation. However, wind tunnels are typically small-scale and may not accurately reflect real-world conditions [23–25]. Despite these limitations, wind tunnels can still provide valuable information about the reduction in airborne particulate matter by vegetation and can be a useful complement to field and computer simulation studies. Overall, a combination of experimental approaches, including wind tunnel studies, can help to provide a comprehensive understanding of the role of vegetation in reducing airborne particulate matter [9,26,27].

When planning and designing hedges for urban areas, landscape architects, agronomists, and urban planners must deal with multiple factors to optimize the positioning of plants, to determine the right botanical species, and to choose the size and density of the vegetal barrier [28]. Elements related to optimization of these parameters are valued in order to minimize planting costs and maximize the ecological benefits of vegetations [29].

The aims of our work were:

- To set up an experimental system that allows for an assessment of the ability of hedges to reduce atmospheric particulate matter;
- To evaluate the relationship between hedge density and dust reduction ability;
- To quantify the air concentration of PM at different distances from green barriers;
- To provide useful elements to design hedges for urban areas.

The experimental setting consisted of a wind tunnel ca. 20 m long and of green barriers made up of mature living laurel plants (*Laurus nobilis* L.). This resulted in an original arrangement which has never been employed before. To our knowledge, this is the first study to be carried out in a wind tunnel of such a size.

## 2. Materials and Methods

In recent years, a work area was created in our institute to evaluate the dispersion of abrasion dust from seeders that simulated the sowing of dressed seeds of maize. Numerous tests were carried out in this facility, and the results were validated through appropriate open field tests [30,31]. The same experimental setup was then utilized in the present study to evaluate the reduction in and the drift of airborne PM incited by a green barrier. The wind tunnel was arranged at the workshop's porch of our institute to obtain a site that was protected by external influences and large enough to contain the hedge and measure dust drift. The test area measured 6.7 m wide, more than 20 m long, and 5.0 m high.

The external open side of the porch was closed by means of tarpaulin sheets settled as curtains (Figure 1). Consequently, the site was closed on each side and the end of the "gallery" was left open to permit the air to flow. At the test site, artificial wind conditions were produced by means of an electric axial fan (0.8 m diameter) operating at 1700 rpm. Preliminary tests were carried out with the aim of verifying the repeatability and the constancy of wind conditions (speed and direction) at the test site by means of a portable anemometer (Schiltknecht, Schaffhausen, Switzerland, Micro MiniAir 4). The wind speed was measured at a height of 0.7 m from the ground at 21 points along the test area.

The diagram in Figure 1 shows the arrangement of the relevant instruments, the source of emissions, the fan, and the green barrier.

The green barrier was made by placing the plants in containers (0.25 m in diameter) side by side to form one, two, or three rows of plants, in order to obtain hedges of different thicknesses and densities (Figure 2). The hedges were placed at 6.0 m from the fan.

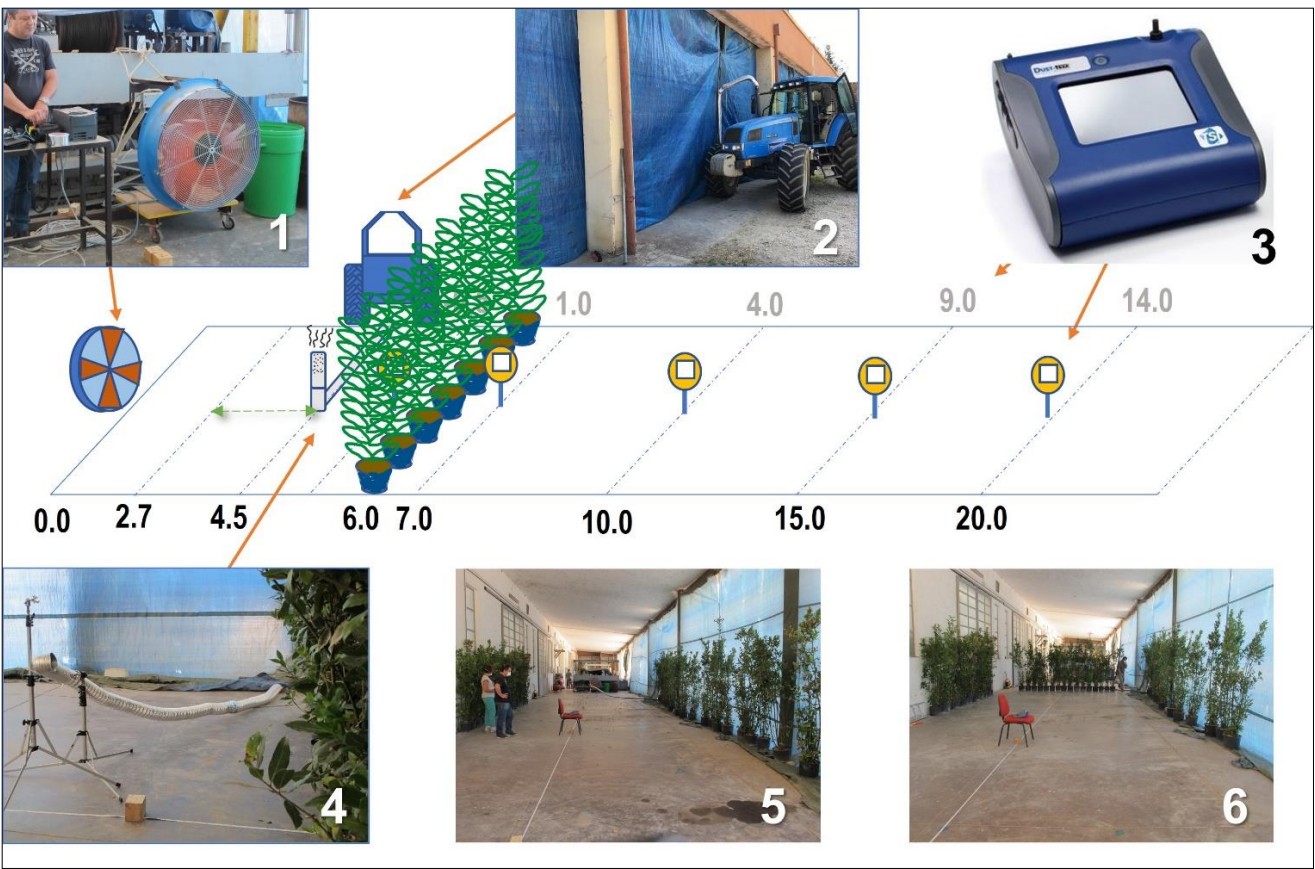

**Figure 1.** Diagram of the experimental area. Numbers are distances (in meters) of the sampling points from the fan (black character) and from the hedge (grey character). (**1**) The fan to generate the air flow; (**2**) the tractor placed outside the wind tunnel with the pipe to collect the exhaust gas; (**3**) the "DustTrak" portable photometer (TSI, Shoreview, MN, USA); (**4**) the point where the exhaust gas of the tractor is released in the test area; (**5**) and (**6**) views of the wind tunnel with and without the hedge.

| | Number of pot rows | Thickness (m) | Height (m) | Width (m) | Porosity (%) | Pots pattern |
|---|---|---|---|---|---|---|
| | 1 | ~ 0.40 | | | 14.3 | |
| | 2 | ~ 0.85 | 2.4 | 5.5 | 4.6 | |
| | 3 | ~ 1.20 | | | 1.4 | |

**Figure 2.** Main characteristics of the hedges.

The optical porosity of the hedge was the parameter chosen to describe its density. This was determined through the analysis of digital images processed with ImageJ software, version 1.53e (Figure 3) [32].

The concentrations of two fractions of PM were then measured: the total suspended particulate (TSP, that is the fraction containing particles with particle diameters <50–100 μm) and the respirable particles ($PM_4$). Measurement of the PM concentration was carried out with a TSI/Tecora "DustTrak" portable photometer capable of detecting, by mounting a suitable cutting head, the concentration of particulate matter (expressed in mg m$^{-3}$) of the

two different size classes. Each acquisition consisted of 30 measures, with a frequency equal to 1 Hz. During sampling, the recorded concentration values were constantly monitored to ensure that data acquisition did not present excessive variability. The sampler operated at a flow rate of 1.7 L min$^{-1}$ for PM$_4$ and 3.0 L min$^{-1}$ for TSP. Before each experiment, the sampling area was carefully cleaned to minimize interferences arising from pre-existing dust.

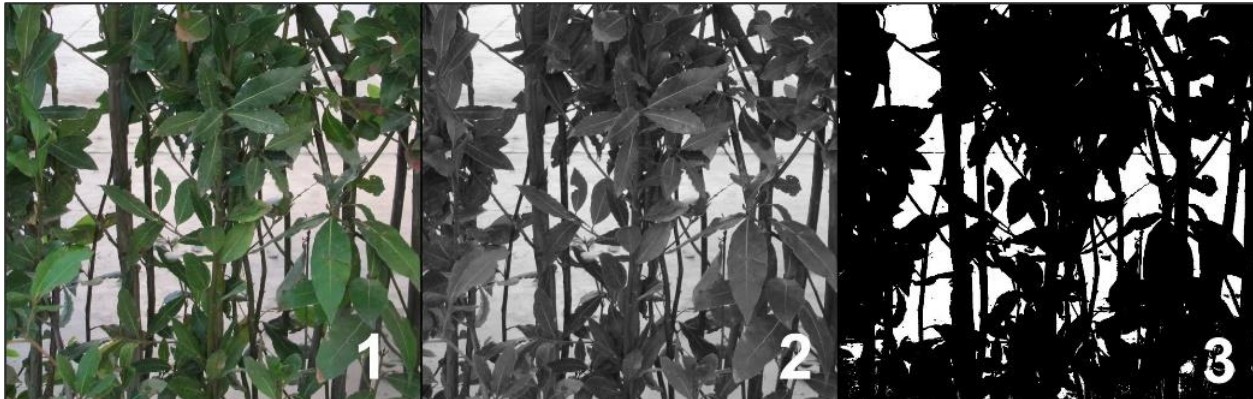

**Figure 3.** Steps of image analysis to estimate the porosity of the hedge. (**1**) Image of the vegetation; (**2**) black and white 8-bit image transformation; (**3**) segmentation and analysis (percentage of the two areas).

Sampling was carried out at 1.0, 4.0, 9.0, and 14.0 m from the hedge, correspondent to 7.0, 10.0, 15.0, and 20.0 m from the fan, at a height of 0.7 m from the ground. Additional samplings of PM were carried in proximity to the gas bypass exit (1.0 m from the outlet) to measure the PM concentration before the hedge.

The source of the particulate matter was the exhaust fumes of a 145 HP Landini Legend tractor (Landini, Reggio Emilia, Italy) operating at 1500 rpm. The tractor was placed outside the wind tunnel, and the exhaust gases were intercepted via a pipe which released them at a height of 0.7 m from the ground, close to the hedge, at 4.5 m from the fan used for the formation of the air flow.

The TSP reduction was also tested in an additional test, with the pipe emitting the exhaust gases placed at 2.7 m from the fan, farther from the vegetation. As mentioned, samplings were repeated at the same points in the absence of the plants and with hedges formed by one, two, and three rows of plants.

In summary, the experimental design entailed PM$_4$ and TSP sampling with 4 hedge configurations at 4 distances from the barrier, and with the fan placed at 2 distances from the hedge. Since each sampling was equal to 30 measures, we obtained a data set of 1820 data points.

Removal efficiency, R, was calculated using the following equation:

$$R[\%] = \frac{PM\ concentration\ without\ plants\ [mg\ m^{-3}] - PM\ concentration\ with\ plants\ [mg\ m^{-3}]}{PM\ concentration\ without\ plants\ [mg\ m^{-3}]} \times 100 \qquad (1)$$

Statistical analysis to determine significant differences between the PM removal capacities of hedges with different densities were performed with a bifactorial ANOVA (analysis of variance) parametric test, with the distance of sampling points and hedge density as factors. Since the variable was not normally distributed (Shapiro–Wilk test, *p*-value < 0.01), it was normalized according to the results of a Box–Cox test procedure, which showed that the best technique of data normalization was the logarithmic transformation. The analyses were performed using R software, version 1.53e [33].

## 3. Results and Discussion

The air velocity inside the wind tunnel sampling area was measured downwind of the hedge, where the average wind speed (measured at a height of 0.7 m from the ground) along the test area was 1.2 m s$^{-1}$ in the absence of the hedge and 0.5 m s$^{-1}$ with the hedge. No substantial differences in average wind velocity were recorded with the hedge formed by one, two, or three rows of plants. The obtained pattern of the air flow is shown in Figure 4, where it is compared the air pattern velocity in the sampling area without plants and with a hedge formed by three rows of pots. The plants induced an average reduction in wind speed, which decreased to zero after approximately 9 m from the hedge. However, around 15 m from the hedge, a slight increase in air velocity was observable (Figure 4B).

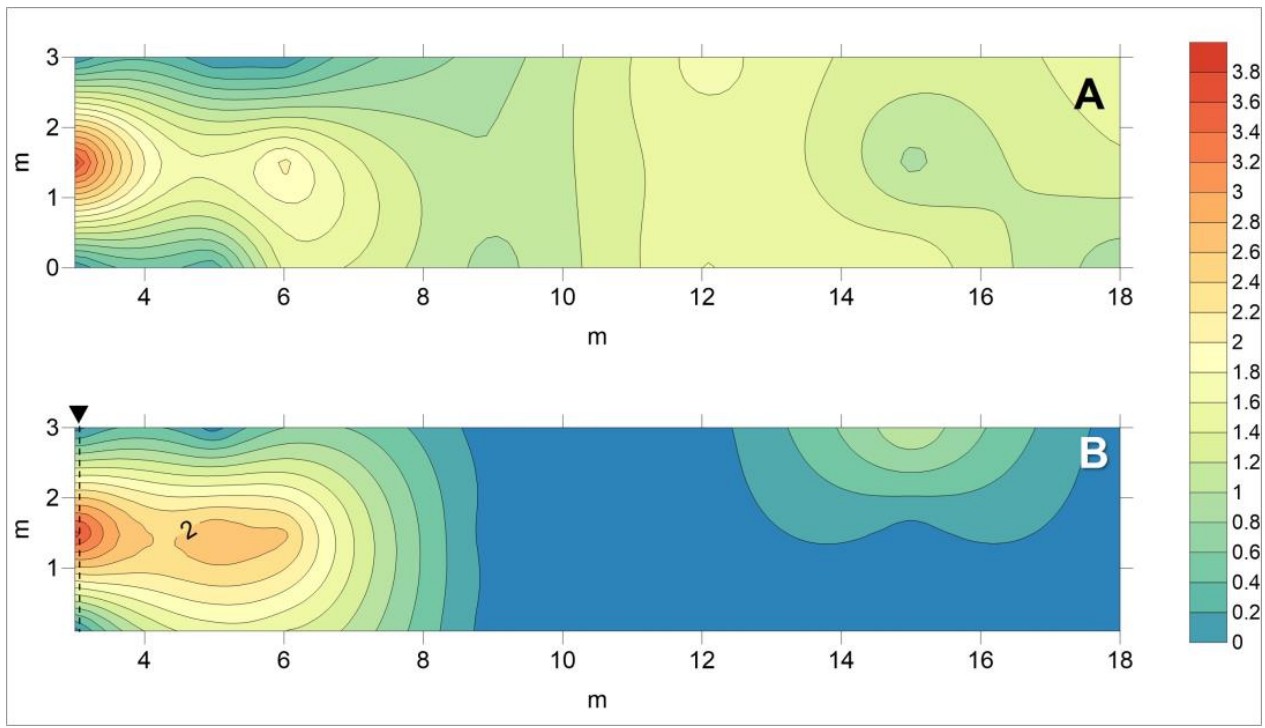

**Figure 4.** Contour maps of air wind patterns in the empty wind tunnel, without plants (**A**) and with a hedge of three rows of plants (**B**). The marked line in (**B**) indicates the positions of plants. Values are expressed in m s$^{-1}$.

The initial concentrations of particles were measured in proximity to the exhaust gas outlet coming from the tractor. The values, reported in Table 1, showed a high concentration of fine particles in the composition of the aerosol.

**Table 1.** Average air concentration values (mg m$^{-3}$ ± standard error) of PM in proximity to the gas exit (the PM$_4$ was not tested at 2.7 m).

| Fraction Size | Distance of the Fan from the Gas Exit | | | | | |
|---|---|---|---|---|---|---|
| | **4.5 m** | | | **2.7** | | |
| TSP | 0.83 | ± | 0.06 | 0.64 | ± | 0.02 |
| PM$_4$ | 4.73 | ± | 0.08 | - | | - |

Regarding the PM reduction detected in the test area, the results are reported in Tables 2–4.

**Table 2.** Reduction in TSP air concentration due to the hedge with respect to the concentration without plants in the test area. Exhaust gas exit was placed at 4.5 m from the fan.

| | Distance of Sampling Points from the Fan (Distance of Sampling Points from the Plants) | | | | |
|---|---|---|---|---|---|
| Row of Plants | 7 (1) m | 10 (4) m | 15 (9) m | 20 (14) m | Average |
| 1 | −9% | −30% | 27% | 39% | 7% |
| 2 | 13% | −3% | 78% | 6% | 24% |
| 3 | 49% | 61% | 3% | 41% | 38% |

**Table 3.** Reduction in $PM_4$ air concentration due to the hedge with respect to the concentration without plants in the test area. Exhaust gas exit was placed at 4.5 m from the fan.

| | Distance of Sampling Points from the Fan (Distance of Sampling Points from the Plants) | | | | |
|---|---|---|---|---|---|
| Row of Plants | 7 (1) m | 10 (4) m | 15 (9) m | 20 (14) m | Average |
| 1 | 21% | 53% | 52% | 32% | 39% |
| 2 | −16% | 82% | 60% | 22% | 37% |
| 3 | 54% | 15% | 44% | 52% | 41% |

**Table 4.** Reduction in TSP air concentration due to the hedge with respect to the concentration without plants in the test area. Exhaust gas exit was placed at 2.7 m from the fan.

| | Distance of Sampling Points from the Fan (Distance of Sampling Points from the Plants) | | | | |
|---|---|---|---|---|---|
| Row of Plants | 7 (1) m | 10 (4) m | 15 (9) m | 20 (14) m | Average |
| 1 | −65% | 48% | 46% | 52% | 20% |
| 2 | 0% | 26% | 23% | 39% | 22% |
| 3 | −31% | 74% | 32% | −84% | −2% |

Descriptive statistics (median, 25th and 75th percentiles, minimum and maximum, outliers) of the dust concentrations in the sampling area for each distance from the hedge are reported in Figure 5. Air concentration values in mg m$^{-3}$ are shown in Table S1 (Supplementary Material). As expected, the PM concentration values in the air correlated with the sampling distance, with partial or no reduction just behind the plants (1 m from the hedge) and maximum reductions in the rest of the sampling area. The ANOVA results, reported in Table 5, showed significant differences according to both of the considered factors, i.e., distance and hedge density.

**Table 5.** Reduction in TSP air concentration due to the hedge with respect to the concentration without plants in the test area. Exhaust gas exit was placed at 2.7 m from the fan.

| Test | Factors | Degrees of Freedom | Probality (*p*-Values) | Significance [1] |
|---|---|---|---|---|
| PM$_4$ | Hedge rows | 3 | <0.001 | *** |
| | Distance | 3 | <0.001 | *** |
| | Hedge rows × Distance | 9 | <0.001 | *** |
| TSP with fan at 4.5 m | Hedge rows | 3 | <0.001 | *** |
| | Distance | 3 | <0.001 | *** |
| | Hedge rows × Distance | 9 | <0.001 | *** |
| TSP with fan at 2.7 m | Hedge rows | 3 | <0.001 | *** |
| | Distance | 3 | <0.001 | *** |
| | Hedge rows × Distance | 9 | <0.001 | *** |

(1)—Significance codes of *p*-values: 0 '***'; 0.001

The maximum overall reductions (41% and 38%) were recorded for $PM_4$ in the case of the thickest hedge (three rows of plants).

A single row of plants can cause a reduction in TSP air concentration only around 9 m from the plant, while in the immediate downwind zone, closer to the hedge, the values are higher. The reduction in $PM_4$, on the other hand, appears to have been more consistent in the various cases.

When the exhaust gas outlet was moved away from the hedge and closer to the fan, the overall TSP reduction pattern significantly changed. In this case, at a distance of 14 m from the hedge, a peak concentration of PM was recorded with the thickest hedge, while the hedges with one or two rows of plants showed a certain abatement effectiveness. As mentioned previously, this set (exhaust gas exit placed at 2.7 m from the fan) was not tested for the finest fraction.

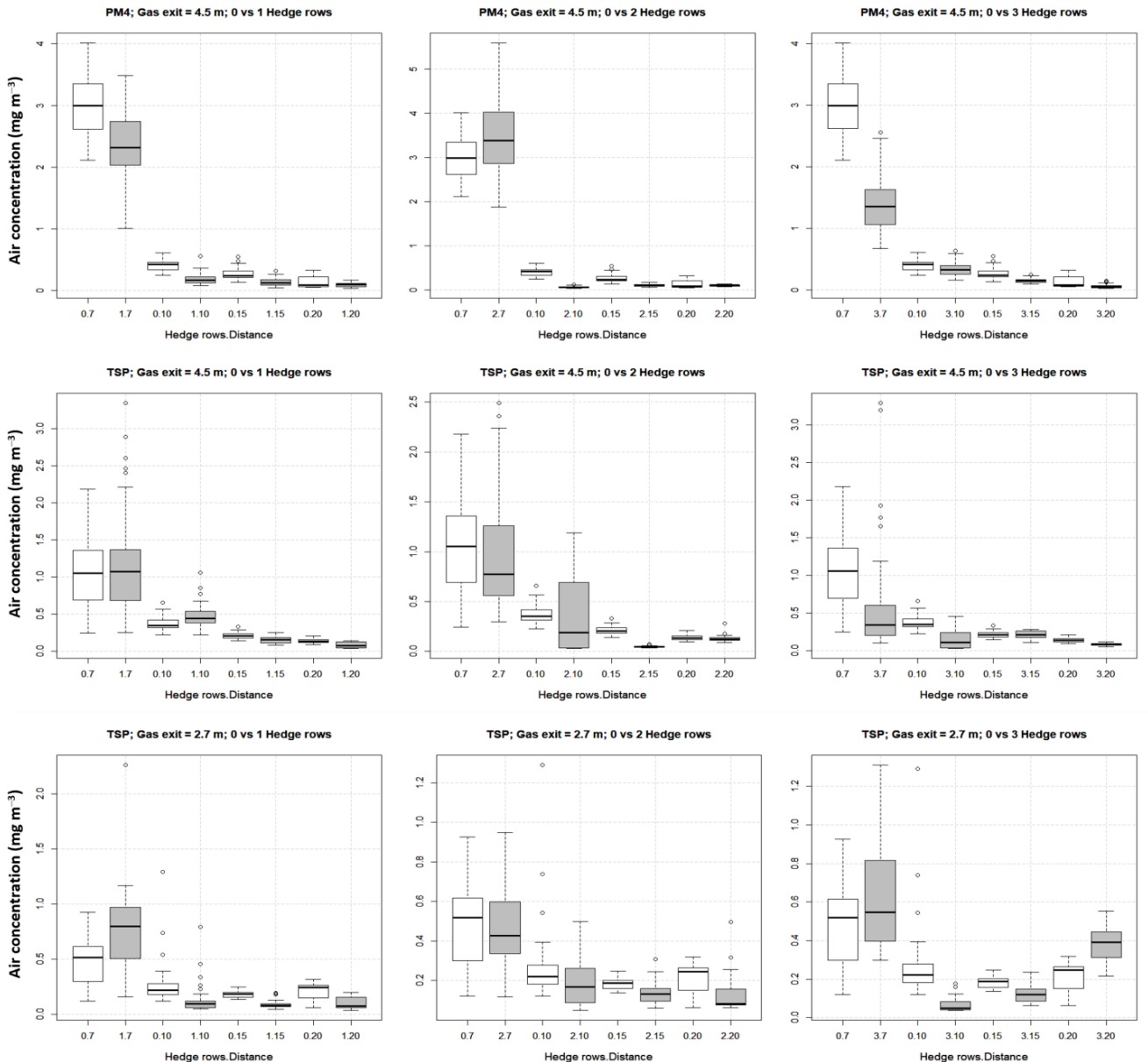

**Figure 5.** Comparison between the values of $PM_4$ and TSP concentrations in the test area, with and without hedges, by the distance. The codes on the horizontal axis show the number of hedge rows and the distances in meters (line within box: median; box limits: 25th and 75th percentiles; whisker ends: minimum and maximum; empty circles: outliers).

The denser screen (three rows), even if, in general, it is more effective in filtering the dust in the various experiments, appeared to be of little effectiveness in the case in which the source of the exhaust gas was moved away from the green screen and closer to the fan (exhaust gas exit placed at 2.7 m from the fan). In this case, it is very likely that the air flow was not able to penetrate the hedge, and thus passed over the hedge. In fact, the wind speed pattern in the sampling area (Figure 4) shows an increase around 15 m. This hypothesis is also supported by the observation that the worst average performance occurred away from the hedge, at the sampling point located at 20 m.

The observed values of overall dust concentration reduction were strictly correlated with the edge porosity (showed in Table 2) in only one test, specifically in the case of TSP assessment with the fan placed at 4.5 m from the exhaust gas exit (Figure 6). In the case of $PM_4$, a relation was not observed because the reduction values were similar with each hedge.

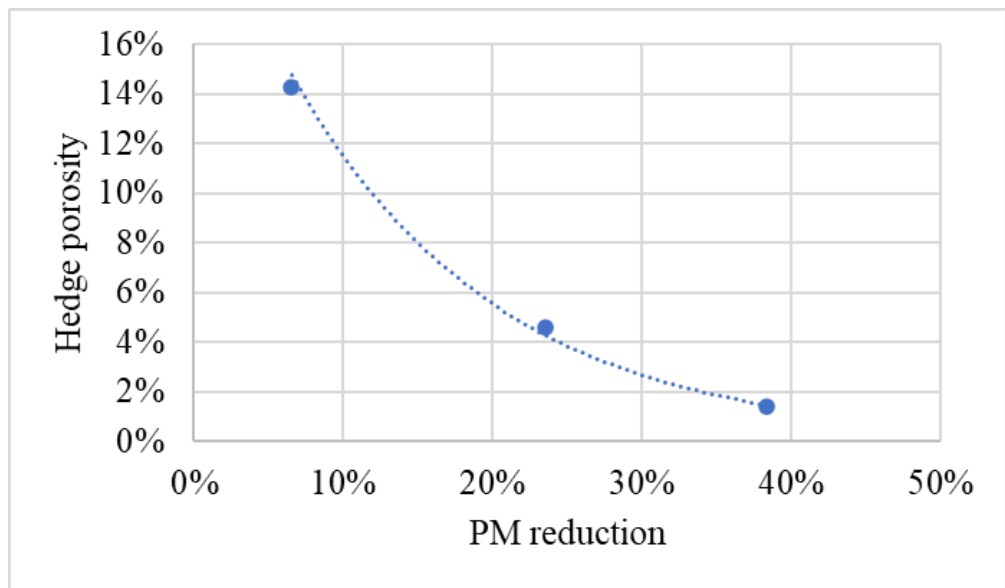

**Figure 6.** Reduction in TSP by the variation in hedge porosity (statistics of the curve: $y = 0.2383 \, e^{-7.289x}$; $R^2 = 0.9987$).

This implies that the total deposition is determined by a trade-off, as stated by Raupach and collaborators [34]: "the windbreak must be dense enough to absorb particles efficiently but sparse enough to allow some particles to flow through and be trapped". Other studies reached similar conclusions thanks to CFD modeling, which highlighted the behavior of particle passing vegetation of different densities in relation to their size [35].

## 4. Conclusions

Wind tunnels are a valuable tool for studying the reduction in airborne particulate matter by vegetation, and many authors have used these facilities. However, wind tunnel experiments may not fully replicate real-world conditions, and the results obtained in a wind tunnel environment may not be translated directly to the field. In particular, small wind tunnels allow for better control of the different factors involved, but the findings are less generalizable. In this study, the size of the wind tunnel allowed for tests of large structures, like those of a real hedge, allowing us to sample the PM up to 15–20 m from the emission source.

We measured the reduction in airborne particulate matter (PM) caused by laurel plants and compared the air concentrations of $PM_4$ and TSP (total suspended particulate) with and without the presence of hedges. On average, reductions in airborne particulate matter concentration were observed with denser screens, but results were also affected by other factors.

We confirmed that the phenomenon was influenced by numerous and complex relations among air movements, vegetation (density, morphology, species, size, etc.) PM concentration, and particle size distribution. The obtained data were affected by multiple involved factors, and for this reason, they often appear very variable. It is likely that, in addition, resuspended dust episodes could affect PM air concentrations independently from the hedge thickness.

A distinctive aspect of this study Is the measurement of dust deposition at varying distances from the green barrier, offering insights into the potential area affected by dust deposition. Such information could be useful in designing and planning hedge placement.

A limit of this study concerns the use of a single plant species, the laurel, a species that is, in any case, widely used to grow hedges in Mediterranean cities.

The comparison of TSP concentrations between the two sets with different distances of the gas exit from the hedge (2.7 and 4.5) showed that denser hedges can act as barriers, which limit the concentration just behind the hedge but have no effect in terms of reducing the overall dust concentration. In this case, dust can jump the hedge and redeposit after some meters. Therefore, the development of a new green structure must consider its density (and dimensions) as a key factor.

In conclusion, it is necessary to underscore the broader significance of urban vegetation in contributing to global sustainability. By enhancing air quality and mitigating the adverse effects of airborne particulate matter, urban vegetation plays a crucial role in fostering a sustainable and resilient environment.

**Supplementary Materials:** The following supporting information can be downloaded at: https://www.mdpi.com/article/10.3390/agriengineering6010014/s1, Table S1: Air concentration values in mg m$^{-3}$.

**Author Contributions:** Conceptualization, M.B. and P.G.; methodology, M.B, G.I. and D.P.; validation, M.B., G.I. and P.G.; investigation, M.B. and P.G.; resources, M.B.; data curation, M.B.; writing—original draft preparation, M.B.; writing—review and editing, M.B.; visualization, M.B.; supervision, M.B. and P.G.; funding acquisition, M.B. All authors have read and agreed to the published version of the manuscript.

**Funding:** This research was funded by the Italian national project URBANFOR3, funded by Lazio Innova (CUP: C82I16000000005).

**Data Availability Statement:** The data presented in this study are available on request from the corresponding author.

**Acknowledgments:** The authors are grateful to Beatrice Bassotti of CREA for her administrative support.

**Conflicts of Interest:** The authors declare no conflicts of interest.

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
