# Peer review of "Reduction in Atmospheric Particulate Matter by Green Hedges in a Wind Tunnel"

_agriengineering, doi:10.3390/agriengineering6010014_

Round 1
Reviewer 1 Report
Comments and Suggestions for Authors
Dear Authors,
I consider this manuscript to be a preliminary study which therefore needs to be improved. Obtained data are affected by multiple factors involved and for this reason they often appear very variable.
Author Response
Reviewer 1
The authors would like to thank the Reviewer for the comments and suggestions. We hope to have addressed below all raised concerns.
Dear Authors,
I consider this manuscript to be a preliminary study which therefore needs to be improved. Obtained data are affected by multiple factors involved and for this reason they often appear very variable.
The manuscript was improved and ameliorated following both the editor’s and the reviewers’ suggestions. I agree that data are affected by multiple factors. This is due to the experimental setting that are similar to a real-world situation.
Reviewer 2 Report
Comments and Suggestions for Authors
1. Abstract. This section mainly describes the experimental results, but more important experimental data should be added.
2. Introduction. What is the novelty of this paper compared to previous studies? This should be further emphasized.
3. Line 183. is showed? This should be checked.
4. Results and discussion. This is the main part of this draft, but the explanation and discussion of the results are lacking. This section needs to be meticulously revised so that the reader can gain something by reading this draft.
5. Figure 6. This experiment needs to be remade. Three points is too little for scientific research, and it is inaccurate.
Comments on the Quality of English LanguageEnglish needs to be revised.
Author Response
Reviewer 2
The authors would like to thank the Reviewer for the comments and suggestions. We hope to have addressed below all raised concerns.
- Abstract. This section mainly describes the experimental results, but more important experimental data should be added.
Additional results and considerations were added to the abstract.
- Introduction. What is the novelty of this paper compared to previous studies? This should be further emphasized.
The novelty of this paper mainly relies on the dimensions of the experimental setting (a large wind tunnel, with real living plants). This point was emphasized in the revised text.
- Line 183. is showed? This should be checked.
The obtained pattern of the air flow is showed in Figure 4,
- Results and discussion. This is the main part of this draft, but the explanation and discussion of the results are lacking. This section needs to be meticulously revised so that the reader can gain something by reading this draft.
This section was ameliorated and expanded. In the tracked version new parts are highlighted.
- Figure 6. This experiment needs to be remade. Three points is too little for scientific research, and it is inaccurate.
I don't easily understand which are the three points you are referring to. The accuracy of the data was ascertained by replicating measurements and carrying out statistical checks.
Comments on the Quality of English Language: English needs to be revised.
The revised manuscript has been proofread by a professional translator.
Reviewer 3 Report
Comments and Suggestions for Authors
Marcello Biocca et al. develop the experimental system and evaluate the ability of the hedges to reduce atmospheric. Potted laurel plants (ca. 2.5 m high) were used to form barriers of different densities and the concentrations of PM in the wind tunnel were compared with and without hedges. They found an overall reduction of air PM concentrations (up to 39%) in the test area in the presence of the hedge. The study could contribute to adding further experimental findings to the complex topic of the interaction between dust and vegetation. However, introduction and discussion are unclear. Before being published, some suggestions should be addressed.
L76-82: Results indicated that different kinds of greenbelts can improve footpath air quality to a certain degree (7 - 15%). Interestingly, the vegetation structure of shrubs and small trees (about 2 - 4.5 m in height) with small crown diameter shows the highest PM10 removal efficiency along major or heavy traffic roads. In open road locations, most studies report a reduction in the concentrations of various airborne pollutants due to hedgerows in the range of 15-60%. Other authors have assessed the pollutant removal capacity in relation to botanical species. However, more studies highlight the the crucial role of biological rhythms. Spring phenology rather than climate dominates the changes in peak of growing season in the Northern Hemisphere (doi.org/10.1111/gcb.16758).
L83-84: Our research focused on the ability of hedges to remove particulate matter or atmospheric dust. There is too little in this paragraph. Please merge with other paragraphs.
L188: Figure 4. Contour maps of air wind patterns in the empty wind tunnel, without plants (A) and with a hedge of three rows of plants (B). Mark the location of the plant.
L230-520: Results and discussion looks like a technical report. Please compare your own results with those of others. Intensify discussion.
Comments on the Quality of English LanguageMarcello Biocca et al. develop the experimental system and evaluate the ability of the hedges to reduce atmospheric. Potted laurel plants (ca. 2.5 m high) were used to form barriers of different densities and the concentrations of PM in the wind tunnel were compared with and without hedges. They found an overall reduction of air PM concentrations (up to 39%) in the test area in the presence of the hedge. The study could contribute to adding further experimental findings to the complex topic of the interaction between dust and vegetation. However, introduction and discussion are unclear. Before being published, some suggestions should be addressed.
L76-82: Results indicated that different kinds of greenbelts can improve footpath air quality to a certain degree (7 - 15%). Interestingly, the vegetation structure of shrubs and small trees (about 2 - 4.5 m in height) with small crown diameter shows the highest PM10 removal efficiency along major or heavy traffic roads. In open road locations, most studies report a reduction in the concentrations of various airborne pollutants due to hedgerows in the range of 15-60%. Other authors have assessed the pollutant removal capacity in relation to botanical species. However, more studies highlight the the crucial role of biological rhythms. Spring phenology rather than climate dominates the changes in peak of growing season in the Northern Hemisphere (doi.org/10.1111/gcb.16758).
L83-84: Our research focused on the ability of hedges to remove particulate matter or atmospheric dust. There is too little in this paragraph. Please merge with other paragraphs.
L188: Figure 4. Contour maps of air wind patterns in the empty wind tunnel, without plants (A) and with a hedge of three rows of plants (B). Mark the location of the plant.
L230-520: Results and discussion looks like a technical report. Please compare your own results with those of others. Intensify discussion.
Author Response
Reviewer 3
The authors would like to thank the Reviewer for the comments and suggestions. We hope to have addressed below all raised concerns.
Comments and Suggestions for Authors
Marcello Biocca et al. develop the experimental system and evaluate the ability of the hedges to reduce atmospheric. Potted laurel plants (ca. 2.5 m high) were used to form barriers of different densities and the concentrations of PM in the wind tunnel were compared with and without hedges. They found an overall reduction of air PM concentrations (up to 39%) in the test area in the presence of the hedge. The study could contribute to adding further experimental findings to the complex topic of the interaction between dust and vegetation. However, introduction and discussion are unclear. Before being published, some suggestions should be addressed.
L76-82: Results indicated that different kinds of greenbelts can improve footpath air quality to a certain degree (7 - 15%). Interestingly, the vegetation structure of shrubs and small trees (about 2 - 4.5 m in height) with small crown diameter shows the highest PM10 removal efficiency along major or heavy traffic roads. In open road locations, most studies report a reduction in the concentrations of various airborne pollutants due to hedgerows in the range of 15-60%. Other authors have assessed the pollutant removal capacity in relation to botanical species. However, more studies highlight the the crucial role of biological rhythms. Spring phenology rather than climate dominates the changes in peak of growing season in the Northern Hemisphere (doi.org/10.1111/gcb.16758).
Thank you for your suggestions. I added information on this point.
L83-84: Our research focused on the ability of hedges to remove particulate matter or atmospheric dust. There is too little in this paragraph. Please merge with other paragraphs.
I isolated this paragraph to underline an objective of the research. Anyway, I've now rewritten the entire section.
L188: Figure 4. Contour maps of air wind patterns in the empty wind tunnel, without plants (A) and with a hedge of three rows of plants (B). Mark the location of the plant.
Ok, the figure was ameliorated.
L230-520: Results and discussion looks like a technical report. Please compare your own results with those of others. Intensify discussion.
The entire section was ameliorated and additional discussion points were added.
Round 2
Reviewer 1 Report
Comments and Suggestions for Authors
Dear Authors,
I'm sorry but I don't think the additions made to the manuscript are sufficient for it to be accepted
Reviewer 2 Report
Comments and Suggestions for Authors
OK
Reviewer 3 Report
Comments and Suggestions for Authors
Accept in present form
Comments on the Quality of English LanguageAccept in present form